# Care home resident identification: A comparison of address matching methods with Natural Language Processing

Víctor Suárez-Paniagua[1,2]*, Arlene Casey[1], Charis A. Marwick[3], Jennifer K. Burton[4], Helen Callaby[5], Isobel Guthrie[6], Bruce Guthrie[1], Beatrice Alex[1,7,8]

1 Usher Institute of Population Health Sciences and Informatics, Advanced Care Research Centre, University of Edinburgh, Edinburgh, United Kingdom, 2 Health Data Research UK, HDR UK, London, United Kingdom, 3 Population Health & Genomics Division, School of Medicine, University of Dundee, Dundee, United Kingdom, 4 School of Cardiovascular and Metabolic Health, University of Glasgow, Glasgow, United Kingdom, 5 NHS Grampian, Aberdeen, United Kingdom, 6 University of St Andrews School of Biology, Biomedical Sciences Research Complex, St Andrews, United Kingdom, 7 School of Literatures, Languages and Cultures (LLC), University of Edinburgh, Edinburgh, United Kingdom, 8 Edinburgh Futures Institute, University of Edinburgh, Edinburgh, United Kingdom

* vsuarez@ed.ac.uk

**Data Availability Statement:** Data cannot be shared publicly to protect patient privacy. The data controllers for the data used in this study are NHS Tayside and NHS Fife, and they require all data

## Abstract

### Background

Care home residents are a highly vulnerable group, but identifying care home residents in routine data is challenging. This study aimed to develop and validate Natural Language Processing (NLP) methods to identify care home residents from primary care address records.

### Methods

The proposed system applies an NLP sequential filtering and preprocessing of text, then the calculation of similarity scores between general practice (GP) addresses and care home registered addresses. Performance was evaluated in a diagnostic test study comparing NLP prediction to independent, gold-standard manual identification of care home addresses. The analysis used population data for 771,588 uniquely written addresses for 819,911 people in two NHS Scotland health board regions. The source code is publicly available at https://github.com/vsuarezpaniagua/NLPcarehome.

### Results

Care home resident identification by NLP methods overall was better in Fife than in Tayside, and better in the over-65s than in the whole population. Methods with the best performance were **Correlation** (sensitivity 90.2%, PPV 92.0%) for Fife data and **Cosine** (sensitivity 90.4%, PPV 93.7%) for Tayside. For people aged ≥65 years, the best methods were **Jensen-Shannon** (sensitivity 91.5%, PPV 98.7%) for Fife and **City Block** (sensitivity 94.4%, PPV 98.3%) for Tayside. These results show the feasibility of applying NLP methods to real data concluding that computing address similarities outperforms previous works.

management and analysis to be carried out in a Safe Haven/Trusted Research Environment (TRE) in accordance with an approved protocol. Researchers can apply to the University of Dundee Health Informatics Centre (HIC) to access the same data via the HIC TRE whose standard operating procedures have been approved by the data controllers (https://www.dundee.ac.uk/hic).

**Funding:** This work was funded by Legal and General PLC (a research grant to establish the independent Advanced Care Research Centre at the University of Edinburgh). The funder had no role in the conduct of the study, interpretation, or the decision to submit for publication. The views expressed are those of the authors and not necessarily those of Legal and General PLC. The work was additionally supported by the Health Data Research UK National Text Analytics Implementation Project, Wellcome Institutional Translation Partnership Awards (PIII029).

**Competing interests:** The authors have declared that no competing interests exist.

## Conclusions

Address-matching techniques using NLP methods can determine with reasonable accuracy if individuals live in a care home based on their GP-registered addresses. The performance of the system exceeds previously reported results such as Postcode matching, Markov score or Phonics score.

## Introduction

Linked routine data is an increasingly powerful tool to understand health and social care needs, patterns of care and outcomes. However, key groups of vulnerable people are relatively invisible in routine data, including residents of care homes [1,2]. This was brutally exposed by the COVID-19 pandemic, where the inability to identify care home residents in routine data seriously constrained pandemic response in a population in which a large proportion of COVID-19 deaths occurred [3–5]. Existing methods for identifying care home residents in routine data typically involve methods which are known to be fallible (although this is rarely quantified) or require considerable amounts of manual matching of addresses [6–8]. Beyond COVID-19, care home residents are a highly vulnerable population with high use of planned and unscheduled healthcare, but the inability to identify them accurately in routine data is a barrier to understanding their needs and improving their care [2].

Identifying care home residents in routine data sources is a challenging task [9,10]. Currently, in the UK identification typically relies on coding in the primary healthcare record (which is very incomplete) or various forms of address matching between the general practitioner (GP) recorded address and the addresses of care homes registered with the regulator.

Natural Language Processing (NLP), a subfield of Artificial Intelligence (AI) has the potential to classify care home residents from their addresses automatically using state-of-the-art Computer Science algorithms. Concretely, NLP methods applied to address matching can be used to extract the relevant information of addresses to check whether a patient belongs to any official care home service.

This study aims to develop NLP methods for identifying care home residents in primary healthcare data and validate their performance in a realistic scenario using a gold standard selected from real population data.

## Literature review

Sherlaw-Johnson et al. [11] examined hospital admission from postcodes known to contain a single care home and observed an unusually high rate of admission from these postcodes, but the analysis included people living in other dwellings with the same postcode and excluded postcodes with more than one care home. Burton et al. [12] explored five methods to classify GP-recorded Community Health Index (CHI) addresses as being care homes with a gold standard sample of 20,000 addresses which had been manually allocated as belonging to a care home or not. Although some of the evaluated methods had excellent performance in identifying care homes in terms of PPV or sensitivity individually, none of them obtained good results in both metrics simultaneously. Schultze et al. [13] compared three methods of identifying care home residents in primary healthcare data (coded as a resident; using address matching; and using the primary care record 'household' identifier to identify households with multiple older residents although this will include sheltered housing residents), with coded data

identifying the fewest and household identifiers the most potential residents, with only partial overlap. However, their study had no gold-standard method for identifying residents and therefore no means of evaluating the accuracy of allocation.

Moreover, Santos et al. [9] proposed using the Ordnance Survey AddressBase Premium database [14] for care home address matching. AddressBase Premium contains all addresses in the UK and assigns a Unique Property Reference Number (UPRN) to each address. The study collected data from people aged 65 years old or over registered by the Care Quality Commission's (CQC) database [15] in England with the same postcode of a care home as potential care home residents. Then, the UPRN matching was validated using a manual address-matched gold standard and obtained promising results, but the study only included 80 English care home addresses out of 15,019 (0.5%), so the feasibility of real population data is unclear. Similarly, the system created by Housley et al. [16] identifies care home residents using postal address matching techniques with CQC registered addresses of care homes in the East Midlands region obtaining good results in the population aged 75 years or over.

Recently, Zhang et al. [17] proposed a machine-learning based model, called FLAP, to link UPRN from free-text addresses that showed a reasonable error tolerance in writing and obtained good accuracy in the experiments. Furthermore, Harper et al. [18] created a deterministic address-matching algorithm, called ASSIGN to allocate UPRN to patient address using gold-standard datasets from London and Wales.

## Methods

### Overall design

The overall design is a diagnostic test study applying and optimizing address-matching methods with NLP to identify care home residents in a training dataset, with a comparison with previous methods [12] in an independent validation dataset. Fig 1 shows the complete pipeline of the proposed method.

### Datasets

The University of Dundee Health Informatics Centre (HIC) provided a list of unique addresses recorded in the NHS Scotland Master CHI registers for residents of NHS Tayside and NHS Fife up to 30th April 2020. CHI addresses are those stored nationally based on information

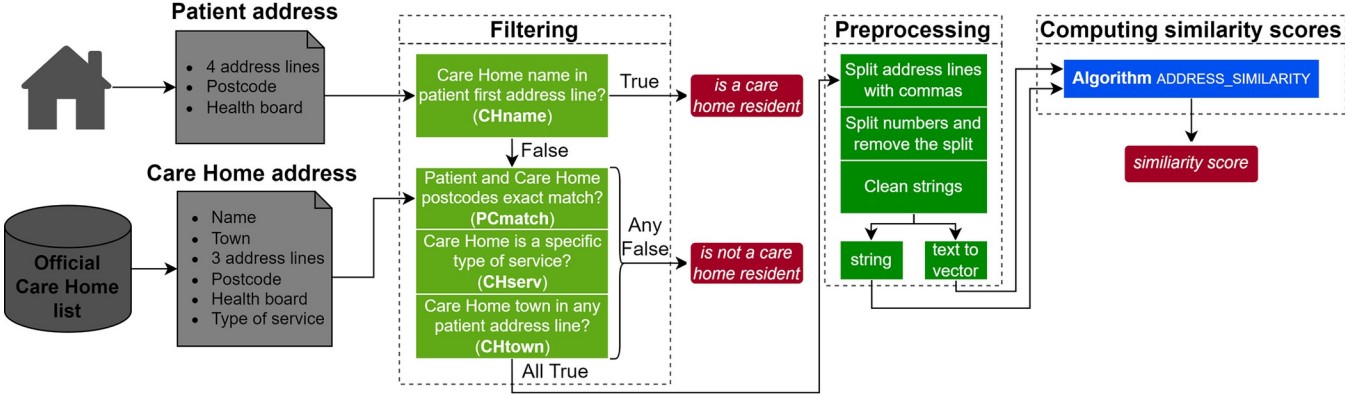

**Fig 1. Address matching system pipeline.** The system takes a patient address and an official care home list as inputs to determine whether is a care home resident or not. The pipeline consists of three main sequential modules: data filtering, data preprocessing, and similarity score computation.

**Table 1. Number of historical and current unique addresses in the CHI and number of people living at a CHI address on 30th April 2020.**

| Population | Unique historical or current addresses in CHI[a] | No. (%) of care home historical or current addresses in CHI[b] | No. of people living at a CHI address on 30/4/20[c] | No. (%) of care home residents on 30/4/20[c] |
|---|---|---|---|---|
| **All ages** | | | | |
| **Fife** | **363,091** | 4,851 (1.3%) | **391,081** | 2,829 (0.7%) |
| Training set (70%) | 254,164 | 3,396 (1.3%) | 273,207 | 1,940 (0.7%) |
| Validation set (30%) | 108,927 | 1,455 (1.3%) | 117,874 | 889 (0.8%) |
| **Tayside** | **408,497** | 5,244 (1.3%) | **428,830** | 3,331 (0.8%) |
| Training set (70%) | 285,949 | 3,671 (1.3%) | 300,448 | 2,492 (0.8%) |
| Validation set (30%) | 122,548 | 1,573 (1.3%) | 128,382 | 839 (0.6%) |
| **Aged ≥65 years** | | | | |
| **Fife** | **60,748** | 608 (1.0%) | **91,575** | 2,685 (2.9%) |
| Training set (70%) | 42,524 | 426 (1.0%) | 64,173 | 1,919 (3.0%) |
| Validation set (30%) | 18,224 | 182 (1.0%) | 27,402 | 766 (2.8%) |
| **Tayside** | **69,475** | 413 (0.6%) | **106,844** | 3,224 (3.0%) |
| Training set (70%) | 48,634 | 290 (0.6%) | 74,732 | 2,378 (3.2%) |
| Validation set (30%) | 20,841 | 123 (0.6%) | 32,112 | 846 (2.6%) |

a. Includes both current and historical addresses (true addresses do not change much over time, but unique addresses recorded in CHI are free-text so the same true address can have multiple different versions in CHI over time).

b. CHI addresses manually identified as care home addresses (gold-standard). This includes historical addresses where no one is resident on 30/4/20.

c. The number of people living at a CHI address on 30/4/20, or the number of people living at a gold-standard care home identified care home address on 30/4/20. Care home addresses are more variable in how they are written than other addresses, and there are large numbers of small care homes for younger people, so for all ages, there are more historical or current care home addresses than current residents.

submitted at the time of registration with a general practitioner (GP) practice for primary health care provision. Patient identifiers were removed from the dataset. The number of residents living at each unique address on 30th April 2020 was also provided, and whether at least one person aged 65 years or over currently lived at that address (a unique address means 'as recorded at GP registration', where the same actual address may have different unique forms, e.g. 10 Glebe Street, Ten Glebe Street, 10 Glebe St. etc.). These datasets were randomly and proportionally split into 70% and 30% for training and validation, respectively (Table 1). Care home addresses were obtained from publicly available data from the Care Inspectorate, the Scottish care regulator, who records care home registered addresses [19].

## Reference standard

Five assessors undertook manual allocation of all CHI addresses to determine if the address was a care home address or not, referring to the Care Inspectorate list of services. A sample of addresses was allocated by multiple assessors, with discussion to achieve consensus where reviewers disagreed. Subsequently, addresses were screened by one assessor, with addresses where there was uncertainty flagged for multiple review and consensus discussion. This manually allocated list was used as the gold standard for this study.

## Data filtering and pre-processing

CHI addresses recorded at GP registration are free-text fields without any required format specification (e.g., the postcode is not required to be in the correct order or appear on its own on a line, and the complete address may be written on one line). Before calculating similarity measures, the datasets were prepared and cleaned using a filtering process and a pre-processing step.

The filtering process had three steps (Fig 1). First, the CHI address was classified as a care home address if the first line contained any phrase related to a care home name such as "care home", "nursing home" or "residential home". Second, the CHI address was classified as *not* being a care home address if the CHI and care home postcodes did not completely exactly match *or* the care inspectorate registered address was for a different kind of care services *or* the CHI address did not include the name of a town contained in any of the care home addresses (S1 Appendix). Third, the remaining unclassified addresses were passed to the pre-processing step.

In the preprocessing step, CHI and care home registered addresses were cleaned as follows: address lines containing commas were split into separated lines, and regular expressions were used to extract numbers together with adjacent letters (excluding spaces and non-alphanumeric characters such as '/'), and all strings were converted to lower case. For example, following these processes, the address line '*89 Bay Street, Newtown-on-Bay*' would be divided into two new address lines and strings cleaned as '[*bay street*, *newtown on bay*]' and a list of numbers '[*89*]'. At the end of this step, the processed address lines are stored as a string or transformed into a vector representation (S2 Appendix).

## Address matching methods

Since both CHI and care home registered addresses are free-text fields, exact address matching is not feasible. Three methods previously applied to these data were applied as baseline comparators: Postcode matching, a Markov method, and a Phonics method [12].

Postcode matching considers a given CHI address as a care home address if its postcode is identical to any postcode in the official care home list. Identifying care home residency in this way is not very accurate overall because the same postcode is often given to different properties. However, this method was validated in previous studies, so we also report it alongside new approaches.

The Markov and Phonics methods create a similarity score where the CHI address is defined as that of a care home if the score is above a defined threshold. In this study, we applied the thresholds proposed in the previous study and (to enable a fair comparison with new methods) optimized thresholds in the training set.

We then applied 11 additional methods for similarity-matching addresses, such as City block, Bray-Curtis, Cosine, Jensen-Shannon, Correlation among others, and optimized them using the training datasets to choose the correct cutoff. All 11 methods calculate a similarity score between addresses (S5 Appendix): four using approximate (or fuzzy) string matching to calculate edit distance (S3 Appendix), and seven using vector space models (S4 Appendix). Also note that References [20–29] are specifically referred to in the appendices.

## Evaluation

The performance of all baseline and new methods was evaluated in terms of positive predictive value (PPV), sensitivity, and the F1 measure. F1 is the harmonic mean of PPV and sensitivity (S6 Appendix). Since being resident in a care home is a relatively rare event within the older adult population, the key performance characteristics desired are high PPV (almost all of those

identified as care home residents truly are residents) and high sensitivity (almost all actual care home residents are identified as residents).

The main objective of this research is to determine which unique addresses are care homes (address level). From a more practical perspective, we can recognize which patients with the resident status can be recognized by knowing the number of people living in each address (patient level). For this reason, the evaluation was carried out at the address level (matching unique addresses) and at the patient level (matching the number of people in each address) using the entire historical dataset on 30th April 2020.

## Results

A total of 771,588 unique CHI current and historical address records for 819,911 people were analysed, of which 408,497 unique addresses for 428,830 were in NHS Tayside and 363,091 unique addresses for 391,081 people were in NHS Fife. Tables 2 and 3 show the performance of the methods for the validation set at the address and patient levels, respectively.

In the evaluation, we carried out different ablation studies and parameter tuning to find the best configuration for the care home address matching system pipeline such as: comparing the filtering processes (S1 Appendix) and the representation of the addresses with the n-gram hyperparameter (S2 Appendix). In addition, we optimized and compared the different edit distance approaches (S3 Appendix), and the vector space models (S4 Appendix) to find the best performance for each population.

**Table 2. Positive predictive value, sensitivity and their harmonic mean F1 for each method of identifying care home at the address level in the CHI address dataset, compared to gold-standard manual identification.**

| Population | Method | PPV | Sensitivity | F1 |
|---|---|---|---|---|
| *All addresses* | | | | |
| Fife | Markov score[a] | 78.0% | 77.3% | 77.6% |
| | City block | 79.5% | 72.1% | 75.6% |
| | **Bray-Curtis** | 83.7% | 77.0% | **80.2%** |
| | Cosine | 85.4% | 70.9% | 77.5% |
| | Jensen-Shannon | 76.7% | 76.0% | 76.3% |
| Tayside | Markov score[a] | 66.5% | 57.2% | 61.5% |
| | City block | 69.6% | 64.6% | 67.0% |
| | Bray-Curtis | 66.8% | 70.4% | 68.6% |
| | **Cosine** | 77.1% | 64.0% | **69.9%** |
| | Jensen-Shannon | 68.8% | 64.1% | 66.4% |
| *Addresses with ≥1 person aged ≥65 years* | | | | |
| Fife | Markov score[a] | 90.9% | 76.4% | 83.0% |
| | City block | 96.7% | 80.2% | 87.7% |
| | Bray-Curtis | 90.5% | 84.1% | 87.2% |
| | Cosine | 92.6% | 82.4% | 87.2% |
| | **Jensen-Shannon** | 96.2% | 83.0% | **89.1%** |
| Tayside | Markov score[a] | 70.0% | 79.7% | 74.5% |
| | **City block** | 94.5% | 83.7% | **88.8%** |
| | Bray-Curtis | 92.5% | 80.5% | 86.1% |
| | Cosine | 95.3% | 82.1% | 88.2% |
| | Jensen-Shannon | 95.2% | 81.3% | 87.7% |

a. Markov score using the same methods as Burton e al [12] but cut-offs fine-tuned for this dataset (allocated as care home when Markov score >47 in Fife all, >62 in Fife ≥65, >37 in Tayside all, >45 in Tayside ≥65).

**Table 3. Positive predictive value, sensitivity and their harmonic mean F1 for each method of identifying care home at the patient level in the CHI address dataset, compared to gold-standard manual identification.**

| Population | Method | PPV | Sensitivity | F1 |
|---|---|---|---|---|
| *All people* | | | | |
| Fife | Markov score[a] | 77.0% | 88.1% | 82.2% |
| | City block | 82.1% | 88.8% | 85.3% |
| | Cosine | 92.6% | 88.5% | 90.5% |
| | Jensen-Shannon | 77.0% | 90.0% | 83.0% |
| | **Correlation** | 92.0% | 90.2% | **91.1%** |
| Tayside all people | Markov score[a] | 71.9% | 89.6% | 79.8% |
| | City block | 72.1% | 92.9% | 81.2% |
| | **Cosine** | 93.7% | 90.4% | **92.0%** |
| | Jensen-Shannon | 70.1% | 92.3% | 79.7% |
| | Correlation | 86.0% | 91.5% | 88.7% |
| *People aged ≥65 years* | | | | |
| Fife | Markov score[a] | 96.7% | 84.2% | 90.0% |
| | City block | 99.0% | 90.2% | 94.4% |
| | Cosine | 97.8% | 91.6% | 94.6% |
| | **Jensen-Shannon** | 98.7% | 91.5% | **95.0%** |
| | Correlation | 98.3% | 91.8% | 94.9% |
| Tayside | Markov score[a] | 91.6% | 86.4% | 88.9% |
| | **City block** | 98.3% | 94.4% | **96.3%** |
| | Cosine | 98.4% | 94.2% | 96.3% |
| | Jensen-Shannon | 98.4% | 94.1% | 96.2% |
| | Correlation | 98.4% | 94.2% | 96.3% |

a. Markov score using the same methods as Burton e al [12] but cut-offs fine-tuned for this dataset (allocated as care home when Markov score >47 in Fife all, >62 in Fife ≥65, >37 in Tayside all, >45 in Tayside ≥65).

We replicated the three previously examined methods [12] finding that the Markov score method performed better than the Phonics score method or the postcode matching, and we compared them with the 11 newly applied methods (S5 Appendix). Therefore, we used the Markov score (with cut-offs from the original study, and cut-offs optimized for this dataset) to compare the four best methods in Tables 2 and 3. General patterns were that: performance was better for Fife data than Tayside data; performance was better in the over-65 population compared to all ages; and the newer methods almost all outperformed the Markov score (fined tuned for this dataset to allow a fair comparison).

The methods which overall performed best at identifying if a CHI address was a care home address (compared to gold-standard manual allocation) for all ages were **Bray-Curtis** (F1 80.2%, sensitivity 77.0%, PPV 83.7%) for the Fife data and **Cosine** (F1 69.9%, sensitivity 64.0%, PPV: 77.1%) for the Tayside data (Table 2). For addresses where at least one person aged ≥65 years lived, the methods with the best performance were **Jensen-Shannon** (F1 89.1%, sensitivity 83.0%, PPV 96.2%) for Fife and **City Block** (F1 88.8%, sensitivity 83.7%, PPV 94.5%) for Tayside. However, differences between methods were relatively small (for example, in the Tayside over-65 population, the worst performing new method was **Bray Curtis** [F1 86.1%, sensitivity 80.5%, PPV 92.5%]).

The methods with the overall best performance for identifying whether an individual was resident in a care home for all ages were **Correlation** (F1 91.1%, sensitivity 90.2%, PPV 92.0%) for Fife data and **Cosine** (F1 92.0%, sensitivity 90.4%, PPV 93.7%) for Tayside (Table 3).

Performance was higher for people aged ≥65 years, with the best methods being **Jensen-Shannon** (F1 95.0%, sensitivity 91.5%, PPV 98.7%) for Fife and **City Block** (F1 96.3%, sensitivity 94.4%, PPV 98.3%) for Tayside. Again, differences between methods were relatively small (for example, in the Fife over-65 population, the worst performing new method was **City Block** (F1 94.4%, sensitivity 90.2%, PPV 99.0%).

## Discussion

### Summary of findings

We validated several address-matching methods to identify care home residents using a large real sample covering the entire population of two health boards in Scotland, NHS Fife and NHS Tayside. Specifically, we developed a pipeline that scores the similarity of a patient address to any of the addresses in a list of Care Inspectorate registered care home services. To do so, we first applied a rule-based filtering process to the GP-recorded addresses in the CHI dataset to deal with special cases, and a preprocessing step using regular expressions to clean the remaining address lines and transform them into a string or vector. Finally, each address was then compared to the official care home list using different similarity-matching methods and classified as a care home address if the computed score was below a pre-trained threshold. We compared three baseline methods proposed in previous work and variations of two types of distance approaches: the edit distance, which quantifies the number of character changes required to transform one address into another, and the vector space model, which transforms each address into a numeric vector. In the experiments, we observed that in general vector space models outperform edit distance ones.

The new methods had higher performance than the best of the baseline methods (the Markov score approach [12]). However, performance varied depending on the population examined. Identification of care home addresses and residents was systematically better in Fife than in Tayside and was systematically better for older people (aged 65 years and over) compared to the whole population. Performance was also better at identifying care home residents than identifying care home addresses reflecting that the addresses with a low number of care home residents are harder to identify.

### Strengths and limitations

This study demonstrates that NLP methods can be effectively applied to address matching to determine if a patient's address is that of a care home. We used a large sample size from two regions of Scotland with mixed city, town, and rural addresses. We tested our methods against a gold-standard researcher-produced dataset. While this study was developed and validated using data extracted on 30[th] April 2020, the presented methods would work on future samples, if the official care home list is up to date to guarantee successful matching between a new CHI address and a care home on service. A key limitation is that the variation in performance between the two regions means that performance may also vary when applied in other parts of the UK, and ideally anyone implementing these methods will validate locally.

### Comparison with other literature

While previous works focused on improving the PPV or sensitivity individually, the present study aimed to obtain the best model that can perform well in both scenarios optimizing the F1 metrics which is its harmonic mean. Compared to the Postcode Matching baseline method also reported in [12], the performance of the best configurations improves from 77.6% to 80.2% F1 using **Bray-Curtis** in Fife at address level (from 82.2% to 91.1% F1 using **Correlation**

at patient level) and from 61.5% to 69.9% F1 using **Cosine** in Tayside at address level (from 79.8% to 92.0% F1 using **Cosine** at patient level). For the population aged 65 or over, F1 increased from 83.0% to 89.1% using **Jensen-Shannon** in Fife at the address level (from 90.0% to 95.0% using **Jensen-Shannon** at the patient level) and from 74.52% to 88.8% using **City Block** in Tayside (from 88.9% to 96.3% using **City Block** at patient level).

## Implications of the study

An integrated health and social care data system would enable care homes to share information about residents moving in and out of their services, which can be linked to other data to build up a picture of their care journey. Address matching of GP-recorded addresses to care home registered addresses is a reasonably effective way of identifying (permanent) care home residents and has the potential to significantly improve our understanding of the needs, service use and outcomes of this vulnerable population. Potential ways of further improving matching would include the use of novel machine learning methods to perform the similarity matching between addresses such as, ensemble methods which use all the different matching scores to classify the resident condition or using state-of-the-art AI methods such as the Siamese Neural Network that can predict the similarity between two addresses [30]. As Santos et al proposed, it may also be more efficient to map both GP-recorded addresses and care home-registered addresses to the Unique Property Reference Number (UPRN) in the Ordnance Survey AddressBase dataset and use UPRN as the linking field. In both cases, the task would be made simpler if GP registration used an online look-up for addresses to ensure that they were recorded in a consistent way (in the way that online shopping does). Alternatively, rather than identifying care home addresses using an inevitably less-than-perfect method, a more systematic recording of who is a care home resident in routine data would be optimal, which will require care homes to securely share these data with public agencies [2].

## Conclusions

The inability to identify care home residents in routine data makes this highly vulnerable population relatively invisible in research and policy. Address matching using NLP methods can identify care home residents with high performance using realistic population data, and in the absence of systematic recording in routine data, can underpin research and understanding for planning and policy.

## Supporting information

**S1 Appendix. Filtering selection.**
(DOCX)

**S2 Appendix. Tuning n-gram.**
(DOCX)

**S3 Appendix. Edit distance.**
(DOCX)

**S4 Appendix. Vector space model.**
(DOCX)

**S5 Appendix. Similarity score computation.**
(DOCX)

**S6 Appendix. Performance measures.**
(DOCX)

# Acknowledgments

The authors would like to thank members of the Clinical Natural Language Processing Research Group and KnowLab at the University of Edinburgh and University College London for their valuable discussion and comments. We are grateful to the University of Dundee Health Informatics Centre for the data provision and technical support.

# Author Contributions

**Conceptualization:** Víctor Suárez-Paniagua, Charis A. Marwick, Jennifer K. Burton.

**Data curation:** Víctor Suárez-Paniagua, Charis A. Marwick, Jennifer K. Burton, Helen Callaby, Isobel Guthrie, Bruce Guthrie.

**Formal analysis:** Víctor Suárez-Paniagua, Arlene Casey, Beatrice Alex.

**Funding acquisition:** Víctor Suárez-Paniagua.

**Investigation:** Víctor Suárez-Paniagua.

**Methodology:** Víctor Suárez-Paniagua.

**Project administration:** Víctor Suárez-Paniagua.

**Resources:** Víctor Suárez-Paniagua.

**Software:** Víctor Suárez-Paniagua.

**Supervision:** Víctor Suárez-Paniagua.

**Validation:** Víctor Suárez-Paniagua.

**Visualization:** Víctor Suárez-Paniagua.

**Writing – original draft:** Víctor Suárez-Paniagua.

**Writing – review & editing:** Víctor Suárez-Paniagua, Arlene Casey, Charis A. Marwick, Jennifer K. Burton, Bruce Guthrie, Beatrice Alex.

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
