## [Decision Letter · Decision Letter 0]

4 Apr 2024

PONE-D-23-42367Care home resident identification: A comparison of address matching methods with Natural Language ProcessingPLOS ONE

Dear Dr. Suárez-Paniagua,

Thank you for submitting your manuscript to PLOS ONE. After careful consideration, we feel that it has merit but does not fully meet PLOS ONE’s publication criteria as it currently stands. Therefore, we invite you to submit a revised version of the manuscript that addresses the points raised during the review process.

We look forward to receiving your revised manuscript.

Kind regards,

Hikmat Ullah Khan, PhD (Computer Science)

Academic Editor

PLOS ONE

Journal Requirements:

5. In the online submission form, you indicated that [The datasets presented in this article are not readily available because they can only be accessed by submitting an application to the information governance body: the Health Informatics Centre of the University of Dundee. Requests to access the datasets should be directed to b.alex@ed.ac.uk.].

6. We notice that your supplementary figures are uploaded with the file type 'Figure'. Please amend the file type to 'Supporting Information'. Please ensure that each Supporting Information file has a legend listed in the manuscript after the references list.

7. We notice that your supplementary tables are included in the manuscript file. Please remove them and upload them with the file type 'Supporting Information'. Please ensure that each Supporting Information file has a legend listed in the manuscript after the references list.

Additional Editor Comments:

In addition to comments and concerns raised by the honorable reviewers, the following comments need the authors ' attention to improve their research work.

Algorithm is poorly represented. It would be better to learn how to write proper pseudocode. Proper symbols to be used. Input and output should be mentioned.

2

Equation number are missing.

3

There are a number of symbols used in the paper. It is recommended to add a Table of symbols sharing the symbols and their description which will serve as a ready reference for the readers.

4

In result section, %age symbols hs too much repetition. Instead it should be used at header of each column once and then repetition with each value will not be required

Reviewers' comments:

Reviewer's Responses to Questions

**Comments to the Author**

1. Is the manuscript technically sound, and do the data support the conclusions?

Reviewer #1: Partly

Reviewer #2: Yes

2. Has the statistical analysis been performed appropriately and rigorously? 

Reviewer #1: I Don't Know

Reviewer #2: No

3. Have the authors made all data underlying the findings in their manuscript fully available?

Reviewer #1: No

Reviewer #2: Yes

4. Is the manuscript presented in an intelligible fashion and written in standard English?

Reviewer #1: No

Reviewer #2: Yes

5. Review Comments to the Author

Reviewer #1: The article needs further arrangement in a research Journal slandered. you should either expand you background of the study more to include discussion on the main topics embedded in the research title like "Care home resident”, “Natural Language Processing" and so on... Or you include a literature review section before the Methodology with more details.

Under the previous study you review also, you need to include details on the methods used by previous researchers, their findings and conclusion of the studies.

Under your method, please also include your algorithm from the appendix to fully show your contributions to the method.

Your framework (Figure 1) is not very clear also.

Reviewer #2: The study used NLP methods (in the preprocess step) for identifying care home residents in primary healthcare data and validated their performance with a manually created ground truth dataset. The paper is a extension of other paper published by some of the here authors.

Generally speaking, the paper is well written.

Some issues:

More information about how the work of the five assessors was executed is required, all five assessors analized all records and a posterior consensus was conduct or a different method was used?

The baseline methods names presented in the result tables should be mentioned in the section "Methods" with the same identification.

No analysis were presented about NPV and Specificity, even though authors argued that the main objective was to optimize the precision and recall measures (throught the F1), I suggest some analysis over the results or a removal from the paper.

The selected method for data split in training and validation (or testing) was the percentage split. Even though it is a applicable method in some contexts, cross-validation is more suitable to ensure that all samples are used at least one time as validation (or testing) data, avoiding a possible bias in the split process.

"While previous works had focused on improving the PPV or sensitivity individually, the present study aimed to obtain the best model that can perform well in both scenarios optimizing the F1 metrics which is its harmonic mean"

As suggestion, the precision-recall curves are a more suitable method to deal with the tradeoff between the precision (PPV) and recall (sensitivity) measures and provide a more holistic threshold independent evaluation usind the Area Under Precision Recal Curve (AUPRC).

6. PLOS authors have the option to publish the peer review history of their article (what does this mean?). If published, this will include your full peer review and any attached files.

Reviewer #1: **Yes: **Mahfooz Ahmed

Reviewer #2: No

---

## [Author Response · Author response to Decision Letter 0]

17 May 2024

the rebuttal letter that responds to each point raised by the academic editor and reviewers has been uploaded as a separate file labeled 'Response to Reviewers'.

Response by Authors #1:

The authors would like to thank the reviewer for the valuable comments and helpful feedback, which have been considered for the revised version of the paper.

The authors have written a completely new and up-to-date section called “Literature review” to expose the background of the current work and the methods used by previous researchers.

In addition, we have aggregated the comparison with previous methods to show the contributions of this new research in the “Comparison with other literature” section.

Moreover, the full algorithm is already made available in the supplementary material, and the authors think that it is best there since only a subset of readers will be interested in reading it. We will of course move it to main text if the editors think that is required.

Furthermore, we have modified Figure 1 to improve the clarity of the proposed method simplifying it and extend the explanation of the system in its caption.

Response by Authors #2:

The authors would like to thank the reviewer for the valuable comments and helpful feedback, which have been considered for the revised version of the paper.

The authors have extended the information on how the five assessors labelled all records in the “Reference standard” section.

In addition, the methods presented in the result table are now mentioned in the “Address matching methods” from the Method section.

Moreover, we have removed the NPV and Specificity metrics from the main text, but we leave them in the tables of the supplementary material for future works.

Furthermore, the authors agree that cross-validation is more suitable to avoid bias in the split process, and that the precision-recall curves and the AUPRC add more information about the results. However, most of the previous studies used PPV, NPV, sensitivity, and specificity over a fixed set, so we wanted to do the same as the related works to compare our results with them on a specific test set and show our improvements over these metrics, therefore we decided to do a detailed study in these metrics and take the F1 measure set to select the best model to use in each scenario.

---

## [Decision Letter · Decision Letter 1]

9 Aug 2024

Care home resident identification: A comparison of address matching methods with Natural Language Processing

PONE-D-23-42367R1

Dear Dr. Suárez-Paniagua,

We’re pleased to inform you that your manuscript has been judged scientifically suitable for publication and will be formally accepted for publication once it meets all outstanding technical requirements.

Kind regards,

Hikmat Ullah Khan, PhD (Computer Science)

Academic Editor

PLOS ONE

Additional Editor Comments (optional):

Reviewers' comments:

Reviewer's Responses to Questions

**Comments to the Author**

1. If the authors have adequately addressed your comments raised in a previous round of review and you feel that this manuscript is now acceptable for publication, you may indicate that here to bypass the “Comments to the Author” section, enter your conflict of interest statement in the “Confidential to Editor” section, and submit your "Accept" recommendation.

Reviewer #2: All comments have been addressed

Reviewer #3: All comments have been addressed

2. Is the manuscript technically sound, and do the data support the conclusions?

Reviewer #2: Yes

Reviewer #3: Yes

3. Has the statistical analysis been performed appropriately and rigorously? 

Reviewer #2: Yes

Reviewer #3: Yes

4. Have the authors made all data underlying the findings in their manuscript fully available?

Reviewer #2: Yes

Reviewer #3: Yes

5. Is the manuscript presented in an intelligible fashion and written in standard English?

Reviewer #2: Yes

Reviewer #3: Yes

6. Review Comments to the Author

Reviewer #2: All my suggestions were answered and partially addressed in the paper. For the cross-validation issue and chosen measures authors presented a plenty justification on their choice of keep as in the original submission.

Reviewer #3: The authors have properly addressed all the comments. The paper is in good enough form and is recommended for acceptance please.

7. PLOS authors have the option to publish the peer review history of their article (what does this mean?). If published, this will include your full peer review and any attached files.

Reviewer #2: No

Reviewer #3: No

---

## [Editor Report · Acceptance letter]

24 Oct 2024

PONE-D-23-42367R1 

PLOS ONE

Dear Dr. Suárez-Paniagua, 

I'm pleased to inform you that your manuscript has been deemed suitable for publication in PLOS ONE. Congratulations! Your manuscript is now being handed over to our production team.

Kind regards, 

on behalf of

Dr. Hikmat Ullah Khan 

Academic Editor

PLOS ONE